# Knowledge of modifiable cardiovascular diseases risk factors and its primary prevention practices among diabetic patients at Jimma University Medical Centre: A cross-sectional study

**Abdata Workina**[1]*, **Asaminew Habtamu**[1], **Tujuba Diribsa**[2], **Fikadu Abebe**[2]

**1** School of Nursing, Jimma University, Jimma, Oromia, Ethiopia, **2** School of Midwifery, Jimma University, Jimma, Oromia, Ethiopia

\* abdeta.15@gmail.com

**Data Availability Statement:** All data are in the manuscript.

**Funding:** The authors received no specific funding for this work.

## Abstract

Cardiovascular diseases (CVDs) are the most common cause of mortality and morbidity globally. This is due to the increasing prevalence of modifiable CVDs risk factors. Hence, the study was aimed to identify knowledge and unhealthy behaviors that contribute to CVD among diabetes mellitus patients. An institutional-based cross-sectional study design was employed among diabetes mellitus patients. A close-ended questionnaire developed from up-to-date similar literature was pretested and face-to-face interview techniques were used to collect data. Checked data were entered into the Epidata 4.1 versions. Then, descriptive and bivariate logistic regression was done using SPSS 25 versions. Of the study participants included in the study, 318, more than half of them, 167(52.5%), were aged ≥45 years and 187(58.8%) of them were females. Among the study participants, more than half, 198 (62.3%), had good Knowledge of modifiable CVDs risk factors. Most of the study participants identified consuming foods rich in fats instead of vegetables and fruits 198(62.3%), followed by physical inactivity 196(61.6%) as a risk factor for CVD. Regarding CVDs prevention practice, 175(55.0%) of the patients had a good practice. More than three-fourths, 267 (84.0%), practice avoiding foods rich in fats and, sugar, and cigarette smoking 250(78.6%). Predictors like educational status, college and above (AOR 2.68; 95% CI 1.14–6.27), and urban residence area (AOR 1.94; 95% CI 1.09–3.15) were associated with knowledge of CVDs risk factors. While sex, marital status monthly income, and age of the participants had no association with knowledge and prevention practice of modifiable CVDs risk factors. The study participants' knowledge and prevention practice of modifiable cardiovascular disease risk factors was satisfactory, even though continuous awareness creation is required to lower CVD mortality and morbidity burdens. Educational status and residence are of the study participants affect the knowledge and prevention practice modifiable of cardiovascular disease risk factors.

**Competing interests:** The authors have declared that no competing interests exist.

## Introduction

CVDs are a group of disorders of the heart and blood vessels including coronary heart disease, cerebrovascular disease(stroke), peripheral arterial disease, rheumatic heart disease, heart defects, hypertension, deep vein thrombosis, and pulmonary embolism [1,2]. Coronary heart disease and stroke were the top two leading causes of CVD health loss in each world region [3].

Cardiovascular diseases are the most common cause of mortality and morbidity globally [4–6]. About 17.9 million people die annually from cardiovascular diseases (CVDs). This represents 32% of all deaths worldwide annually. Of these deaths >75% happen in middle- and low-income countries [7]. More than 22.2 million people will die annually by 2030 due to CVDs unless intervened [8].

The modifiable CVD risk factors include smoking tobacco, hypercholesterolemia, diabetes, sedentary lifestyle, overweight/obesity, high-fat content diet, and excessive alcohol consumption [3,6,9]. According to, 2017 global burdens of disease report, high blood pressure, and smoking were the leading global risk factors causing early death and disability for all age groups [10].

In Ethiopia, dyslipidemia (90.6%), physical inactivity (76%), and hypertension (62.7%) were the most common prevalent cardiovascular disease risk factors [11].

The effects of behavioral risk factors that contribute to the development of cardiovascular diseases and its complication may show up in individuals as raised blood pressure, raised blood glucose, raised blood lipids, and obesity. Furthermore, economic and cultural change, globalization, urbanization, population aging, poverty, and hereditary factors were the underlying determinants of CVDs [12,13].

Most cardiovascular diseases can be prevented by cessation of tobacco use, reduction of salt in the diet, eating more fruit and vegetables instead of eating high-fat content foods, regular physical activity, and avoiding excessive alcohol consumption have been shown to reduce the risk of cardiovascular disease. Creating conducive health policy environments that making healthy choices affordable and available are essential for motivating people to adopt and sustain healthy behaviors [3,12–14].

Additionally, early detection and treatment adherence of hypertension, diabetes and high blood lipids are a cost-effective approach to reduce cardiovascular disease burdens both in high and low-income countries [15].

Different studies were conducted regarding knowledge and prevention practice of modifiable cardiovascular disease worldwide, even though the global pandemic of modifiable cardiovascular disease risk factors were as usual especially in low-income countries [5,6,15–19]. However, to date, no information was found in Ethiopia regarding knowledge and prevention practice of modifiable cardiovascular disease. Hence, this study was aimed to identify knowledge and unhealthy behaviors that contributed to CVDs and factors associated with it.

## Patients and methods

### Ethics statement

The ethical clearance approved by Jimma university institute of health IRB with Ref No: IHRPGD/469/2020 was obtained and given to Jimma University's medical centre administrator. Verbal informed consent was obtained from the hospital administrator to collect the data and this study was conducted in accordance with the Helsinki declaration. Written informed consent was taken from each study participant. Data collectors explained the objectives, their right to refuse and discontinue the data collection. Participant's name was not recorded as well as their confidentiality was highly secured through this study process.

## Study design and setting

An institutional-based cross-sectional study design was conducted at Jimma university medical center (JUMC) from January 8, 2021, to February 24, 2021. Jimma university medical center was located in the southwest of Ethiopia, providing teaching and research services in addition to medical services. The medical center had provided services for more than 20 million patients with 800 beds from the southwest of Ethiopia [20,21].

## Study population

Patients with diabetes mellitus visited Jimma university medical center for follow-up at the diabetic clinic during the data collection period (January 8, 2021, to February 24, 2021).

## Study variables

The predictor study variables were sociodemographic characteristics (age, sex, marital status, occupation, educational status, residence area, monthly income and types of DM and dependent variables were knowledge and prevention practice of modifiable CVDs risk factors.

## Eligibility criteria

Diabetic Mellitus patients whose age was 18 years old and above who visited the study site for follow-up were included in the study while diabetic mellitus patients who have already developed cardiovascular disease and mentally ill were excluded from the study.

## Sample size determination and sampling procedure

The required sample size was determined using single population formula with the assumption of confidence interval 95%, the margin of error 5%, and since no study was conducted in Ethiopia on knowledge of cardiovascular disease risk factors and its primary prevention practice proportion of diabetic patients who had good knowledge of cardiovascular disease risk factors (p) 50% was considered; initial sample size (ni) = $(Z\alpha/2)2p (1-p)/d2$ = 384. Since the sample size was taken from a population of $< 10, 000$ the initial sample size was adjusted using the correction formula; final sample size (nf) = ni (N)/ ni+N = 294, where N = 1260 number of diabetic mellitus patients who were on follow up at JUMC. Finally, after adding a 10% non-response rate a total of 323 diabetic patients were targeted for the study. The list of patient's record orders on the appointment chart was used as a sampling frame. To select each study unit, the systematic sampling technique was employed using the sampling fraction (k) = N/n; 1260/323 = 4. Then each study participant was selected by adding a sample fraction until reaching the total sample size targeted for this study and the first study participants were selected using lottery methods.

## Data collection tool and procedure

A closed-ended questionnaire was developed to update similar literature which contains socio-demographic characteristics, knowledge of modifiable cardiovascular disease risk factors, and cardiovascular disease risk factors prevention measuring variables [18,19,22–25]. Knowledge of modifiable CVDs risk factors and its prevention practices were dichotomized into good or poor knowledge of cardiovascular disease risk factors or prevention practices of cardiovascular disease. Good knowledge of cardiovascular disease risk factors or primary prevention practices of cardiovascular diseases were computed from the mean score and those participants who had knowledge of CVDs risk factors or its prevention practices above mean

score were considered to have good knowledge or good CVDs prevention practice [16,22,26]. Data were collected by 5 BSc nurses through face-to-face interview techniques.

## Data quality assurance

The questionnaire was pretested on 5% of the sample size on patients with diabetes mellitus who were on follow-up at Shenen Gibe hospital. During the pre-test, each variable of the questionnaire was assessed for its understand-ability, sensitivity, and reliability statistics was computed with Cronbach's alpha of 0.83. The questionnaire was translated from the English version to Afan Oromo and Amharic languages by language experts for data collection then, back to English during data analysis. Data collectors obtained training three days before data collection and the supervisor provided on-site close supervision and checked the completeness of the questionnaires' during data collection.

## Data processing and analysis

Checked data were entered into Epi-data entry client 4.6 versions. Then, cleaned data were exported and analyzed using SPSS 25.0 versions. Descriptive statistics were used to summarize categorical variables of patients' socio-demographic characteristics, knowledge of risk factors, and prevention practice of CVD. The knowledge of CVD risk factors and its prevention practice was dichotomized into good/poor knowledge of CVD risk factors or CVD prevention practice then, it was analyzed using binary logistic regression. The fit of the model was checked by the Hosmer-Lemeshow goodness of fit test. In the binary logistic regression predictors with p-value < 0.25 at 95% CI were candidates for multivariate logistic regression. In the multivariate logistic regression independent variables with a p-value of < 0.05 were considered statistically significant association.

## Results

### Socio-demographic characteristics

Of the study participants included in the study, 318, more than half of them, 167(52.5%), were aged ≥45 years and 187(58.8%) of them were females. Regarding educational status, most of them, 95(29.9%) were educated up preparatory school followed by college and above 81 (25.5%) whereas, 66(20.8%) of the study participants were illiterate. Concerning residential areas, 210(66.0%) of the study participants reside in urban areas. Among the study participants, 249(78.3%) had type 2 diabetes mellitus, while the rest had type 1 diabetic mellitus (**Table 1**).

### Knowledge of modifiable cardiovascular disease risk factors

Amongst the study participants, 198(62.3%) of them were identified consuming foods rich in fats instead of vegetables and fruits might cause CVD, followed by physical inactivity 196 (61.6%) while, only less than half, 152(47.8%), the patients knew that cigarette smoking was a risk factor of cardiovascular diseases. More than half, 198(62.3%), of the study participants had good Knowledge of modifiable CVDs risk factors (**Table 2**).

### Primary prevention practices of cardiovascular diseases

Among the study participants more than three fourth, 267(84.0%), practice avoiding foods rich in fats, sugar, and salt, followed by avoiding cigarette smoking 250(78.6%), then adherence to DM treatment protocol 237(74.5%) although, only 137(43.1%) of them practice reduce

**Table 1. Socio-demographic characteristics of patients with diabetic mellitus who were on follow-up at Jimma University Medical Centre, 2021.**

| Variables(n = 318) | | Frequency | Percent (%) |
|---|---|---|---|
| Age (in yrs.) | <45 | 151 | 47.5 |
| | ≥45 | 167 | 52.5 |
| Sex | Male | 131 | 41.2 |
| | Female | 187 | 58.8 |
| Marital status | Married | 219 | 68.9 |
| | Single | 60 | 18.9 |
| | Divorced | 25 | 7.9 |
| | Widowed | 14 | 4.4 |
| Occupation | Farmer | 60 | 18.9 |
| | Merchant | 68 | 21.4 |
| | Government employee | 51 | 16.0 |
| | Other* | 56 | 17.6 |
| | Housewife | 83 | 26.1 |
| Educational status | Elementary | 76 | 23.9 |
| | High school/preparatory | 95 | 29.9 |
| | College and above | 81 | 25.5 |
| | Illiterate | 66 | 20.8 |
| Residence area | Urban | 210 | 66.0 |
| | Rural | 108 | 34.0 |
| Monthly income (in ETB) | = >5000 | 38 | 11.9 |
| | <5000 | 280 | 88.1 |
| Types of DM patients had | Type 2 DM | 249 | 78.3 |
| | Type 1 DM | 69 | 21.7 |

Note

* pensioner, student, and jobless.

alcohol consumption. Concerning cardiovascular diseases prevention practice status, 175 (55.0%) of the patients had a good practice (Table 3).

## Factors associated with knowledge of modifiable cardiovascular diseases risk factors and its primary prevention practices

To identify factors associated with knowledge of modifiable cardiovascular disease risk factors and its prevention practices, knowledge of modifiable CVDs and its prevention practices was dichotomized into good or poor knowledge of risk factors and good or poor practice of CVD prevention. Then a cross tab was computed with predictor variables to identify whether cells were sufficient to perform logistic regression. Model fit was checked by using Hosmer and Lemeshow's goodness of fit test. In the bivariate logistic regression predictors with p-value < 0.25 were candidates for multivariate logistic regression.

In the bivariate logistic regression, age ≥ 45 years (COR 1.31; 95% CI .83–2.06), occupation of the patient, merchant (COR 2.93; 95% CI 1.42–6.08), government employee, (COR 5.70; 95% CI 2.36–13.77), housewife (COR 1.14; 95% CI .58–2.21), elementary (COR 2.14; 95% CI 1.15–3.95), high school/preparatory (COR 6.58; 95% CI 3.22–13.48), college and above (COR 5.07; 95% CI 2.44–10.50) and urban residence (COR 2.77; 95% CI 1.71–4.48) were factors associated with knowledge of modifiable CVDs risk factors, while sex and marital status of the participants had no association with knowledge of CVDs risk factors (**Table 4**).

**Table 2. Knowledge of modifiable cardiovascular diseases risk factors among diabetic mellitus patients who were on follow-up at Jimma University Medical Centre, 2021.**

| Variables | | Frequency | Percent (%) |
|---|---|---|---|
| Is high blood pressure a risk factor for CVD? | Yes | 174 | 54.7 |
| | No | 54 | 17.0 |
| | I don't know | 90 | 28.3 |
| Is overweight a risk factor for CVD? | Yes | 181 | 56.9 |
| | No | 71 | 22.3 |
| | I don't know | 66 | 20.8 |
| Is excessive alcohol intake a risk factor for CVD? | Yes | 169 | 53.1 |
| | No | 76 | 23.9 |
| | I don't know | 73 | 23.0 |
| Does DM be a risk factor for CVD? | Yes | 153 | 48.1 |
| | No | 71 | 22.3 |
| | I don't know | 94 | 29.6 |
| Is physical inactivity a risk factor for CVD? | Yes | 196 | 61.6 |
| | No | 42 | 13.2 |
| | I don't know | 80 | 25.2 |
| Is cigarette smoking a risk factor for CVD? | Yes | 152 | 47.8 |
| | No | 71 | 22.3 |
| | I don't know | 95 | 29.9 |
| Is consuming foods rich in fats instead of vegetables and fruit a risk factor for CVD? | Yes | 198 | 62.3 |
| | No | 72 | 22.6 |
| | I don't know | 48 | 15.1 |
| Knowledge of modifiable CVDs risk factors | Poor knowledge | 120 | 37.7 |
| | Good knowledge | 198 | 62.3 |

Based on multivariate logistic regression those participants who had education status of college and above were 2.7 times more likely to have good knowledge of CVDs risk factors than illiterate participants (AOR 2.68; 95% CI 1.14–6.27) and additionally, urban residence area (AOR 1.94; 95% CI 1.09–3.15) were associated with knowledge of CVDs risk factors (Table 4).

**Table 3. Primary prevention practices of cardiovascular diseases among diabetes mellitus patients who were on follow-up at Jimma University Medical Centre, 2021.**

| Variables (n = 318) | | Frequency | Percent (%) |
|---|---|---|---|
| Avoiding foods rich in fats, sugar, and salt | Yes | 267 | 84.0 |
| | No | 51 | 16.0 |
| Regular physical activity | Yes | 198 | 62.3 |
| | No | 120 | 37.7 |
| Adherence to DM treatment protocol | Yes | 237 | 74.5 |
| | No | 81 | 25.5 |
| Avoid cigarette smoking | Yes | 250 | 78.6 |
| | No | 68 | 21.4 |
| Reduce alcohol consumption | Yes | 137 | 43.1 |
| | No | 181 | 56.9 |
| Screening for high blood pressure | Yes | 142 | 44.7 |
| | No | 176 | 55.3 |
| CVDs prevention practice status | Poor practice | 143 | 45.0 |
| | Good practice | 175 | 55.0 |

**Table 4. Factors associated with knowledge of modifiable cardiovascular diseases risk factors and its primary prevention practices among diabetic mellitus patients who were on follow-up at Jimma University Medical Centre, 2021.**

| Variables | | Good knowledge of CVDs risk factors | | | Good practice of CVDs prevention | | |
|---|---|---|---|---|---|---|---|
| | | P-value | COR (95% C.I) | AOR (95% C.I) | P-value | COR (95% C.I) | AOR (95% C.I) |
| Age in yr. | ≥45 | .249 | 1.31(.83–2.06) | | .045 | 1.58(1.01–2.47) | |
| | <45 | | 1.00 | | | 1.00 | |
| Sex | Male | .298 | 1.28(.80–2.04) | | .261 | .77(.49–1.21) | |
| | Female | | 1.00 | | | 1.00 | |
| Marital status | Married | .511 | 2.32(.78–6.92) | | .007* | .59(.19–1.82) | .10(.03-.39) |
| | Single | | 2.15(.66–6.98) | | | .78(.23–2.60) | .08(.02-.36) |
| | Divorced | | 2.37(.62–9.03) | | | 1.43(.35–5.79) | .37(.08-.84) |
| | Widowed | | 1.00 | | | 1.00 | |
| Occupation | Farmer | 0.004 | 1.00 | | 0.011 | 1.00 | |
| | Merchant | | 2.93(1.42–6.08) | | | 2.41(1.18–4.91) | |
| | Government employee | | 5.70(2.36–13.77) | | | 6.53(2.70–15.82) | |
| | Housewife | | 1.14(.58–2.21) | | | 1.02(.52–2.00) | |
| Educational status | Illiterate | .001* | 1.00 | | .000* | 1.00 | |
| | Elementary | | 2.14(1.15–3.95) | 1.73(.90–3.32) | | 4.71(2.36–9.44) | 7.82(3.38–18.09) |
| | High school/preparatory | | 6.58(3.22–13.48) | 4.05(1.82–9.01) | | 9.11(4.37–19.01) | 16.98(7.03–41.02) |
| | College and above | | 5.07(2.44–10.50) | 2.68(1.14–6.27) | | 16.58(7.24–37.98) | 30.28(11.61–78.98) |
| Residence area | Urban | 0.010* | 2.77(1.71–4.48) | 1.94(1.09–3.15) | .045 | 1.61(1.01–2.57) | |
| | Rural | | 1.00 | | | 1.00 | |
| Monthly income | ≥5000 ETB | .236 | .64(.31–1.34) | | .016 | .40(.19-.84) | |
| | <5000 ETB | | 1.00 | | | 1.00 | |

Note

*p-value <0.05 in multivariate logistic regression.

Abbreviations: COR, crude odds ratio; AOR, adjusted odds ratio; CI, confidence interval.

With regard to CVDs prevention practices, marital status of married (AOR.10; 95% CI.03-.39), single (AOR.08; 95% CI (.02-.36), and divorced (AOR .37; 95% CI .08-.84) and educational status of; elementary (AOR 7.82; 95% CI 3.38–18.09), high school/preparatory (AOR 16.98; 95% CI 7.03–41.02), college and above (AOR 30.28; 95% CI 11.61–78.98) were predictors of good CVDs prevention practices (**Table 4**).

## Discussion

CVDs are the leading cause of premature death and disability worldwide [6,13,13,27]. Prevention of modifiable cardiovascular disease (CVD) risk factors have significantly reduced CVD mortality and morbidity [28]. Since magnitude of CVDs risk factors in the populations change over time and need continuous awareness creation and other intervention, this study was aimed to show gap on knowledge and primary prevention practice of modifiable cardiovascular disease risk factors and associated factors and might be used as secondary source of data for researchers who want to research on the same inquiry.

This study shows that more than half, 198(62.3%), of the study participants had good Knowledge of modifiable CVDs risk factors and most of them were identified consuming foods rich in fats instead of vegetables and fruits 198(62.3%), physical inactivity 196(61.6%) and hypertension 174 (54.7%), while only less than half of the study participants knew,

cigarette smoking 152 (47.8%), and DM 153(48.1%) was a risk factor of CVDs. This study finding was relatively consistent with a study conducted in Malaysia which shows, 58% of the participants know hypertension is a risk of CVD, [26] and as well with the study conducted in Egypt, which shows 88(44%) of the participants mentioned high blood pressure is a risk factor for heart diseases [25]. This finding can robust on evidence of key knowledge of CVDs risk factors similarity over various countries.

Whereas, it was lower than a study conducted in Nigeria among health care workers which revealed 86.2% of the participants had good knowledge of chronic heart disease risk factors [29]. Again another study conducted in Nigeria among university students shows that smoking 342(85.1%), and hypertension 334(83.1%), was a risk for developing heart disease [24]. This discrepancy could be related to a study conducted in Nigeria, in which the study participants were health workers and university students with higher knowledge regarding CVDs. However, our study finding was higher than the study conducted in Tanzania revealed, only 25.4% of the participants had good knowledge of CVD risk factors [30]. This inconsistency might be due to the researchers were used open-ended questions to assess the knowledge of CVD risk factors and the participants included in the study were only rural residential. Similarly, this study finding was higher than another study finding that revealed, most participants had unsatisfactory knowledge of risk factors (64.6%) [31].

This study shows that amongst study participants 175(55.0%) of the patients had a good CVD prevention practice. More than three fourth, 267(84.0%), of them, practice avoiding foods rich in fats, sugar, and salt, followed by avoiding cigarette smoking 250(78.6%), then adherence to DM treatment protocol 237(74.5%) while, only 137(43.1%) of them practiced reducing of alcohol consumption. This study finding was relatively consistent with the study conducted in Ibadan, 54.8% of the participants had good preventive behaviour against CVDs, [24] and Nigeria, which shows, 114 (75.0%) of the participants eat fruits and vegetables regularly and 101 (66.4%) of them ensure appropriate treatment of diabetes mellitus [29]. Moreover, our study finding was consistent with another study conducted in Nigeria that spectacles, 81 (48.2%) had good practice of primary CVD prevention [15]. This shows that there almost related behavioural practices of modifiable cardiovascular disease risk factors.

However, our study finding was higher than the study conducted in Nepal and Kenya that spectacle, 113 (31.7%) and 27(9.0%) of the respondents screened regularly for high blood pressure respectively [18,23]. This disparity could be due to differences in healthcare availability and study participants across studies. On the other hand, our study was lower than other undertaken in Iraq that shows, 82.5% of participants reported that they had their blood pressure checked [19].

Based on multivariate logistic regression educational status and residence area had a strong association with knowledge modifiable CVDs risk factors, college and above (AOR 2.68; 95% CI 1.14–6.27), and (AOR 1.94; 95% CI 1.09–3.15), respectively. While there was no association with sex, age, income, and marital status of study participants. This study finding was similar to the study conducted in Cameroon, which shows a high level of education (AOR = 2.26 (1.69–3.02), p = 0.0001) was associated with good knowledge [32]. Furthermore, our study was consistent with studies conducted in Nigeria, Iraq, and Kenya that show there was no significant difference in knowledge scores among gender of participants, no significant association was found between the participants' gender, marital status, and knowledge level (P > 0.05), and higher education was a strong predictor of CVD risk factor knowledge (6.72, 95% CI 1.98–22.84, P < 0.0001) respectively [18,33,34]. This consistency shows that participants who had higher educational level might research out about their health. So, creating awareness could be tackled unhealthy behaviors of cardiovascular diseases.

On the other hand, our study was in contrast with another two studies undertaken in Nigeria and Palestine which reveals, sex and age, and gender, age-group (p = 0.039), and gender (p = 0.0.19) of participants were associated with knowledge of CVDs risk factors respectively [24,35,36]. This discrepancy might be due to sociodemographic characteristics difference among countries of studies.

## Strengths and limitations of the study

The study's shortcoming was because it was focused on a single institution, generalization as a whole may not have been considered and since the duration of data collection was short it might affect the finding of the result.

The strengths of this study were that there is no information regarding knowledge and primary prevention practice of modifiable CVDs risk factors which makes the study more significant. Furthermore, the study participants were even at risk of developing the disease. So, the study can identify whether the study site institution provided information about unhealthy behaviors of CVDs and continuously monitored it.

## Conclusion

The study participants' knowledge of modifiable cardiovascular disease risk factors was satisfactory, even though continuous awareness creation is required to lower CVD mortality and morbidity burdens. Primary cardiovascular diseases prevention practice among the study participants was good, but needs to be improved more.

Those patients who had higher educational level and residing in the urban area were more knowledgeable on modifiable CVDs risk factors and its prevention practice, whereas age, sex, marital status and income level of the participant had no effect on knowledge and prevention practice of modifiable CVDs risk factors.

## Acknowledgments

We would like to thank Jimma University medical center for technical support during data collection and our study participants for their willingness to be interviewed.

## Author Contributions

**Conceptualization:** Abdata Workina.

**Data curation:** Fikadu Abebe.

**Formal analysis:** Asaminew Habtamu.

**Software:** Abdata Workina.

**Validation:** Tujuba Diribsa.

**Writing – original draft:** Abdata Workina.

**Writing – review & editing:** Tujuba Diribsa, Fikadu Abebe.

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
