## [Decision Letter · Decision Letter 0]

9 Feb 2022

PGPH-D-22-00130

Knowledge of Modifiable Cardiovascular Diseases Risk Factors and Its Primary Prevention Practices among Diabetic Patients at Jimma University Medical Centre: A cross-sectional study

Dear Dr. Workina,

Thank you for submitting your manuscript to PLOS Global Public Health. After careful consideration, we feel that it has merit but does not fully meet PLOS Global Public Health’s publication criteria as it currently stands. Therefore, we invite you to submit a revised version of the manuscript that addresses the points raised during the review process.

We look forward to receiving your revised manuscript.

Kind regards,

Guglielmo Campus, Ph.D DDS

Academic Editor

Journal Requirements:

1. Please amend your Financial Disclosure statement. If you did not receive any funding for this study, please simply state: “The authors received no specific funding for this work.”

2. Please update your Competing Interests statement. If you have no competing interests to declare, please state: “The authors have declared that no competing interests exist.”

3. In the online submission form, you indicated that “The datasets generated and analysed during the current study are not publicly available due to these data were used under license for the current study but are available from the corresponding author on reasonable request.”. All PLOS journals now require all data underlying the findings described in their manuscript to be freely available to other researchers, either 1. In a public repository, 2. Within the manuscript itself, or 3. Uploaded as supplementary information.

4. Please ensure that the Title in your manuscript file and the Title provided in your online submission form are the same.

Additional Editor Comments (if provided):

Reviewers' comments:

Reviewer's Responses to Questions

**Comments to the Author**

1. Does this manuscript meet PLOS Global Public Health’s publication criteria? Is the manuscript technically sound, and do the data support the conclusions? The manuscript must describe methodologically and ethically rigorous research with conclusions that are appropriately drawn based on the data presented.

Reviewer #1: Partly

Reviewer #2: Partly

2. Has the statistical analysis been performed appropriately and rigorously?

Reviewer #1: Yes

Reviewer #2: Yes

3. Have the authors made all data underlying the findings in their manuscript fully available (please refer to the Data Availability Statement at the start of the manuscript PDF file)?

Reviewer #1: Yes

Reviewer #2: No

4. Is the manuscript presented in an intelligible fashion and written in standard English?

Reviewer #1: Yes

Reviewer #2: Yes

5. Review Comments to the Author

Reviewer #1: A good study but the conclusion needs to be re-worded. The second paragraph of the discussion reads "...This study shows that more than half, 198(62.3%), of the study participants had good Knowledge of modifiable CVDs risk factors..."; therefore the conclusion that "-----The study participants' knowledge of modifiable cardiovascular disease risk factors was insufficient", does not seem congruent since majority (over half) of respondents have good knowledge

Rephrase the last paragraph of the conclusion (typographical errors too).

Reviewer #2: Study analysis of care for modifiable cardiovascular disease risk factors and their primary prevention practices among diabetic patients at Jimma University Medical Center: Cross-sectional study, the authors present after a relevant and current topic.

By following the reading in the Summary and Introduction Sections as a manuscript of some elements, which I missed in these, reading in the comments of the file as suggestions for changes.

In general, as the study presents, some are mentioned in the title “primary prevention”, with many important examples of reference, as well as important in introduction in this aspect, both in the discussion sections.

The study also needs to be more detailed and described in the methodological part, as well as better developed in the Discussion section. In the end, the Discussion was not carried out, there is information and results that must be discussed, studied and justified at the end of the conclusion that does not present the fact, it is proposed in the initial and objective hypothesis. In fact, the objective can be revised, as I believe that knowledge and practices cannot be solved in a single study and carried out in such a short time with a not so substantial sample. The Conclusion of the study is presented as a part that can be developed in the Discussion of the same, in my opinion.

It is important to pay attention to the ability of individuals to seek, understand, evaluate and make sense of health information, that is, health literacy and its dimensions. In this sense, ignoring sociodemographic and economic factors for sample selection may, in this case, bias the final result in terms of the responses acquired in the study. Without distinction of relative or absolute population, how to affirm results without having the same perspective of comparison? Suggestion: review, analyze, clarify...

In the Discussion section, several important topics can and should be addressed as the fundamental aspect that relates the health professional/services offered and their commitment to the development of skills and competences for self-care in health. After all, who is responsible for non-good health practices? And what are the factors that influence these practices and/or absences from them and why? Finally, the article should show readers what was the purpose of showing the knowledge of such practices and the importance of this for the health area and those directly involved (individuals with heart disease/family members, health professionals and services) in the cardiovascular diseases theme.

Below are comments and suggestions for revision and corrections.

I congratulate the authors once again for choosing the topic and for the work carried out. I recommend publishing after making the suggested adjustments and corrections.

I would like to have access to the article after pre-publication authors' adjustment.

Thank you for the opportunity to learn and exchange knowledge.

Cordially, good work and success to all!

Comments/Suggestions

Page 4

Suggestion: insert the period in which the survey took place, as well as the variables

Page 5

This study shows that more than half, 198(62.3%), of the study participants had good Knowledge of modifiable CVDs risk factors and in Results, 175(55.0%) of the patients had a good CVD prevention practice, and if 55.5%, that is, more than half of the population studied had good practices, it is risky to state in the conclusion that "practices are insufficient"... Bias!? Suggestion: review!

Page 6

Suggestion: relocate this paragraph as first.

Page 7

Little study time. Why? Suggestion: look for other similar studies to verify the research time issue. Justify!

Page 8

What are the issues? How many?

Page 9

Suggestion: information must be in the Abstract

Page 11

If this study shows that more than half, 198(62.3%), of the study participants had good Knowledge of

modifiable CVDs risk factors and 55.5%, that is, more than half of the population studied had good practices, it is risky to state in the conclusion that "practices are insufficient"... Bias!? Suggestion: review!

Page 12

Why this value and not 0.5? Suggestion: justify!

Page 15

Suggestion: discuss the results found and compared studies. Discuss the main points justifying and giving perspectives... It lacked to justify the importance and the reason of the study carried out as a whole.

Page 16

And what is the conclusion of this? Suggestion: develop what has been compared...discuss such facts! Remove data, percentages.

Page 17

Suggestion: I would also add the study time as a limiting factor, as well as the calculation of the listed population.

6. PLOS authors have the option to publish the peer review history of their article (what does this mean?). If published, this will include your full peer review and any attached files.

**Do you want your identity to be public for this peer review?** For information about this choice, including consent withdrawal, please see our Privacy Policy.

Reviewer #1: No

Reviewer #2: No

---

## [Decision Letter · Decision Letter 1]

16 Jun 2022

Knowledge of Modifiable Cardiovascular Diseases Risk Factors and Its Primary Prevention Practices among Diabetic Patients at Jimma University Medical Centre: A cross-sectional study

PGPH-D-22-00130R1

Dear Mr Workina,

We are pleased to inform you that your manuscript 'Knowledge of Modifiable Cardiovascular Diseases Risk Factors and Its Primary Prevention Practices among Diabetic Patients at Jimma University Medical Centre: A cross-sectional study' has been provisionally accepted for publication in PLOS Global Public Health.

Best regards,

Maurizio Trevisan, M.D., MS

Academic Editor